# Gene expression profiling identifies candidate biomarkers for new latent tuberculosis infections. A cohort study

Mariana Herrera[1,2], Yoav Keynan[1,3,4], Paul J. McLaren[1,5], Juan Pablo Isaza[6], Bernard Abrenica[5], Lucelly López[6], Diana Marin[6], Zulma Vanessa Rueda[1,6] *

**1** Departments of Medical Microbiology & Infectious Diseases, University of Manitoba, Winnipeg, Canada, **2** Doctorado en Epidemiologia, Facultad Nacional de Salud Pública, Universidad de Antioquia, Medellín, Colombia, **3** Department of Internal Medicine, University of Manitoba, Winnipeg, Canada, **4** Department of Community Health Sciences, University of Manitoba, Winnipeg, Canada, **5** JC Wilt Infectious Diseases Research Centre, Public Health Agency of Canada, Winnipeg, Manitoba, Canada, **6** Facultad de Medicina, Universidad Pontificia Bolivariana, Medellín, Colombia

\* zulma.rueda@umanitoba.ca

**Data Availability Statement:** The raw data of the RNA-sequencing is on the SRA repository. Bioproject: New Mycobacterium tuberculosis infections. A cohort study. Accession:

## Abstract

### Objective

To determine the gene expression profile in individuals with new latent tuberculosis infection (LTBI), and to compare them with people with active tuberculosis (TB) and those exposed to TB but not infected.

### Design

A prospective cohort study. Recruitment and follow-up were conducted between September 2016 to December 2018. Gene expression and data processing and analysis from April 2019 to April 2021.

### Setting

Two male Colombian prisons.

### Participants

15 new tuberculin skin test (TST) converters (negative TST at baseline that became positive during follow-up), 11 people that continued with a negative TST after two years of follow-up, and 10 people with pulmonary ATB.

### Main outcome measures

Gene expression profile using RNA sequencing from PBMC samples. The differential expression was assessed using the DESeq2 package in Bioconductor. Genes with |logFC| >1.0 and an adjusted p-value < 0.1 were differentially expressed. We analyzed the differences in the enrichment of KEGG pathways in each group using InterMiner.

PRJNA858854; ID: 858854. URL: https://www-ncbi-nlm-nih-gov.uml.idm.oclc.org/bioproject/PRJNA858854/.

**Funding:** This research was funded by the Departamento Administrativo de Ciencia, Tecnología e Innovación (COLCIENCIAS), currently named as Ministerio de Ciencia, Tecnología e Innovación (Minciencias), in the form of the grant "Host gene expression profile used to identify latent TB infection and the transition to active disease - Perfil de la expresión génica del hospedero para identificar tuberculosis latente y la transición a enfermedad activa" [grant no. 121071249878] and by a National Ph.D. [grant no. 067C-04/18-55] to ZVR and by the Centro de Investigación para el Desarrollo y la Innovación, Universidad Pontificia Bolivariana, as part of the project "Análisis de expresión génica y su correlación con la terapia antituberculosa," [grant no. 832B-06/17-55] in the form of funds to DM. This study was also funded by the Departamento Administrativo de Ciencia, Tecnología e Innovación (COLCIENCIAS), currently named as Ministerio de Ciencia, Tecnología e Innovación (Minciencias), in the form of a doctoral scholarship in Colombia [program #647] and a Emerging Leaders of Americas Program, 2018, doctoral scholarship from the Government of Canada to MHD. This research was also supported in part by a Canada Research Chairs Program for ZVR [award # 323473], specifically to cover the open access fee and the salary during the manuscript writing and editing process. The funders had no role in study design, data collection and analysis, decision to publish, or preparation of the manuscript.

**Competing interests:** The authors have declared that no competing interests exist.

## Results

The gene expression was affected by the time of incarceration. We identified group-specific differentially expressed genes between the groups: 289 genes in people with a new LTBI and short incarceration (less than three months of incarceration), 117 in those with LTBI and long incarceration (one or more years of incarceration), 26 in ATB, and 276 in the exposed but non-infected individuals. Four pathways encompassed the largest number of down and up-regulated genes among individuals with LTBI and short incarceration: cytokine signaling, signal transduction, neutrophil degranulation, and innate immune system. In individuals with LTBI and long incarceration, the only enriched pathway within up-regulated genes was Emi1 phosphorylation.

## Conclusions

Recent infection with MTB *is* associated with an identifiable RNA pattern related to innate immune system pathways that can be used to prioritize LTBI treatment for those at greatest risk for developing active TB.

## Introduction

Tuberculosis (TB) remains a leading infectious disease of public health concern worldwide, with about a third of the global population infected by Mycobacterium tuberculosis [1] (MTB). Individuals with latent TB infection (LTBI) serve as a reservoir for the bacterium, are at risk of progressing to active TB, and therefore pose a risk of spreading the infection to their families and the community. To control TB, it is essential to prevent new cases of TB, and one of the ways is to offer treatment for LTBI [2, 3].

Given the high number of people infected by the MTB, offering massive treatment, although a highly desirable option, is not feasible for low- and middle-income countries where most of those infected reside [4]. Therefore, diagnosing and treating patients with new MTB infection within the first months after the infection could be a promising strategy to prevent progression to active TB in people with LTBI, because they are at the highest risk for progression to active TB [1, 5, 6].

Identifying those recently infected is one of the main challenges because it can only be done in prospective studies. Existing tests for the diagnosis of LTBI, i.e. the tuberculin skin test (TST) and interferon-γ release assays (IGRAs), are unable to predict the time of infection, cannot distinguish cleared infection from persistent infection, have low sensitivity in some populations, and cannot differentiate between LTBI and active TB, among other previously published disadvantages [1, 7–13]. For this reason, it is essential to identify new targets for diagnostic tests or improve the available tests for diagnosing LTBI.

In recent years, several published studies have reported the use of RNA sequencing in different types of samples, to predict the progression of infected individuals to active TB [14, 15], to predict the outcome at the time of completion of anti-TB treatment [16], to identify differential gene expression profiles between active TB, LTBI, and non-infected people [17, 18], to distinguish LTBI from healthy individuals [19] and to differentiate people with pulmonary TB, extrapulmonary TB and other lung infectious diseases [20, 21]. Few cross-sectional studies have been conducted to identify LTBI exclusively, and those studies have focused on detecting micro RNAs [19, 22], and the different clinical stages of people diagnosed with LTBI [23].

Studying individuals with a new LTBI can better understand the earliest events in the interaction between the host and the pathogen, including the host's immune response to mycobacteria at the earliest time of infection. This approach might also allow the identification of a specific gene expression profile that can identify candidate biomarkers that enable the detection of people with recent infection and offer a treatment aimed at groups with the highest risk of progression to active TB. Therefore, this study aimed to determine the gene expression profile in individuals with new MTB infection and compare them with people with active TB and those exposed to mycobacteria but not infected.

## Methods

### Ethics considerations

Approval for the cohort study was obtained from the Ethics Committees of the Universidad Pontifica Bolivariana (July 15, 2015) and the University of Manitoba (HS19804 (H2016:218). The Instituto Nacional Penitenciario y Carcelario (INPEC) and the director of each prison approved the project.

We have been working in prisons since 2010. A detailed process of how we approach People deprived of liberty (PDL) has been previously published [24]. Briefly, the field team (two nurses) visited the prisons from Monday to Friday and obtained written informed consent. They explained the project, the benefits, risks, samples, tests, etc., and invited the PDL to participate. We gave the PDL the written consent form, which was explained and signed in the presence of two witnesses. These witnesses were PDL, and they also signed the written consent form. We never took the consent form in the presence of security guards to avoid undue pressure. After this process, we collected sociodemographic and clinical information, administered the tuberculin skin test, and took the blood samples. At all times, the PDL could ask questions and refuse to participate. During the follow-ups, all documents containing information that could identify a participant were code protected.

TB and LTBI treatment: All PDL diagnosed with active TB received treatment according to the National guidelines, independently of their acceptance to participate in the study. New converters were reported to the prison health authority in both prisons, and PDL from prison 1 were offered LTBI treatment. LTBI diagnosis and treatment in PDL are not considered mandatory in the international and Colombian guidelines; the healthcare personnel from prison 2 opted not to offer LTBI treatment. Therefore, new converters incarcerated in that facility did not receive LTBI therapy.

Inclusivity in global research: Additional information regarding the ethical, cultural, and scientific considerations specific to inclusivity in global research is included in the Supporting Information (S1 File).

### Study design and population

We conducted a cohort study between September 2016 and December 2018 in two male prisons in Antioquia, Colombia [25]. Fig 1 shows the cohort design, the number of prospectively enrolled and followed and the number of processed samples. The inclusion criteria are described in the S2 File. Participants with a two-step TST negative at baseline were tested for TB infection every six months during the study period using TST. TST was administered based on the CDC recommendations [26].

During the follow-up, we identified 99 individuals who remained TST negative and 25 who converted to TST positive. A converter was defined as TST$\geq$10 mm with an increase of at least 6 mm between the measure of TST at baseline and a new TST during the follow-up [27].

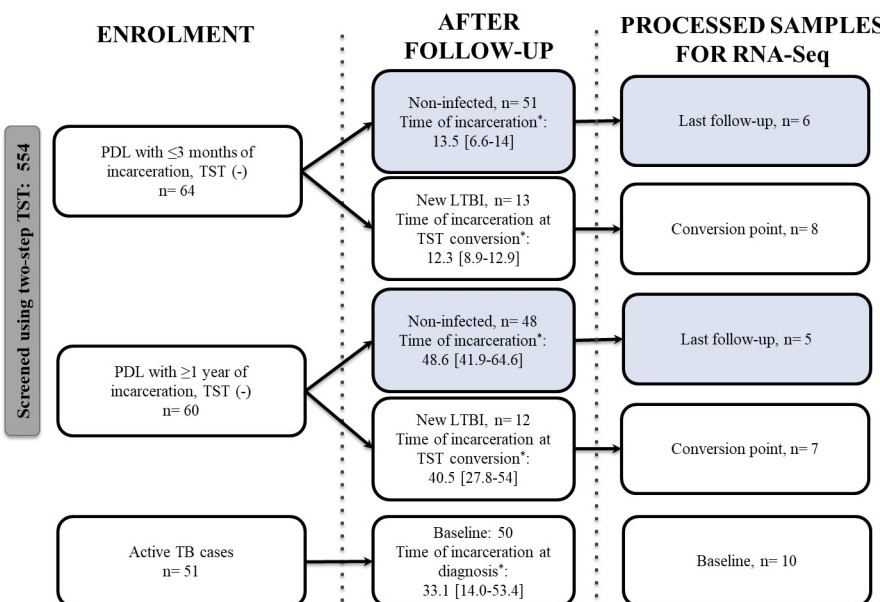

**Fig 1. Flow chart of people included in the cohort study and the outcome at the end of the follow-up.** PDL: People deprived of liberty. TB: tuberculosis; TST: tuberculin skin test; LTBI: latent tuberculosis infection. *Months, Median [IQR].

During the cohort study, we found that the tuberculosis infection incidence rate varied between 2,402.88 cases per 100,000 person-months (95% CI 1,364.62–4,231.10) in PDL with short incarceration (those who were enrolled in follow-up upon incarceration or within the first three months of incarceration), to 419.66 cases per 100,000 person-months (95% CI 225.80–779.95) in individuals with long incarceration (PDL who started their follow-up after one year or more of incarceration) [25]. Based on this finding, the time of incarceration is a variable that may affect the risk of becoming infected and sick. Therefore, we divided individuals with and without new LTBI, with short incarceration [SI] or long incarceration [LI].

For RNA sequencing, we selected samples from individuals with HIV-negative results that had the longest time of follow-up:

- **11 non-infected people**: PDL that at baseline had two-step negative TST and never converted their TST during follow-up (time of follow-up, median [IQR]; 15.3 [13.0–43.9] months). Among non-infected people, there were 6 individuals with a short time of incarceration (NI-SI, median 13.4 months [12.7–14.7]), and 5 with a long time of incarceration (NI-LI, median 43.8 months [28.7–56.8]).

- **15 new TST converters**: PDL that at baseline had two-step negative TST and then converted their TST during follow-up, without an active TB diagnosis. Among the converters, 8 had a new LTBI and SI (LTBI-SI, median [IQR] 12.1 months [10–12.5]), and 7 had a new LTBI and LI (LTBI-LI, median 39.4 months [24.2–60.3]).

- **10 patients with pulmonary TB**: TB microbiologically confirmed (culture and auramine-rhodamine stain), with less than 5 days of TB treatment. Time of incarceration until the diagnosis of active TB: median [IQR]; 34.1 [15–69] months. The diagnosis criteria are shown in the S2 File.

## Sociodemographic variables

After entry to the cohort and during follow-up, we collected the following variables: age, history and time of prior incarceration, use of drugs (inhaled, injected, or smoked) or alcohol, comorbidities (chronic obstructive pulmonary disease, diabetes, chronic kidney disease), contact with an active TB case (outside and inside the prison), weight and height to calculate body mass index (BMI), and if at any time they develop respiratory symptoms; and BCG vaccine status checked by the presence of a BCG scar.

## Procedures

**Sample.** Blood samples were collected at baseline from all the participants and every three months until the end of the follow-up in people with LTBI and non-infected people. PBMCs were separated by density gradient with Ficoll (Sigma-Aldrich, Missouri, US), and preserved at -121ºC in DMSO solution, bovine fetal serum, and RNA later® (Ambion, Texas, US) until RNA extraction.

**RNA extraction.** Samples were thawed at 37ºC for 2 min and separated by centrifugation at 5000g/5min. Total RNA extraction was done from the PBMCs using the commercial kit RNeasy® Plus Mini Kit (Qiagen, Hilden, Germany), with the following modifications: 10 min of incubation in the lysis buffer and treatment on column with RNase-Free DNase Set (Qiagen, Hilden, Germany) for 15 min. The total RNA was suspended in 30ul of RNase-free water (Qiagen, Hilden, Germany). RNA quality was evaluated using the TapeStation (Agilent Technologies, Inc, California, US), RNA concentration was assessed with Quibit 2.0 (Thermo Fisher, Massachusetts, US), and purity was evaluated using Nanodrop™ 2000c (Thermo Fisher, Massachusetts, US). Samples with an RNA Integrity Number (RIN) ≥7, the ratio of 260/280 ≥1.8 and 28S/18S ratio ≥1.5 were selected for sequencing. The evaluations and subsequent steps followed established protocols by Sheng Q *et al.* [28] and Conesa A *et al.* [29].

**Library preparation and RNA sequencing.** The cDNA libraries generated paired-end reads and were prepared using TruSeq stranded mRNA library kit (Illumina, California, US), and the sequencing was done by Macrogen Inc. (Seoul, Korea) using the NovaSeq system (Illumina, California, US).

**RNA-seq data analysis.** We performed quality control using FastQC (Babraham Bioinformatics) [30], and the adapters and low-quality sequences (i.e. Phred score <30) were trimmed by Trimmomatic program v0.36 [31]. When all sequences met the quality criteria established for RNA in the FastQC program, the filtered reads were aligned to the reference *Homo sapiens* genome GRCh38 [32] using HiSATv2.1.0 [33]. Then, aligned reads were quantified to obtain the gene-level counts using featureCounts v1.6.2 [34].

## Analysis

Our primary outcome was new LTBI, both in PDL with a short and long time of incarceration.

Initially, we assessed the changes in gene expression of PBMC among people with a new LTBI, active tuberculosis and non-infected. After that, we evaluated the changes in gene expression between PDL with new LTBI with a short time of incarceration (LTBI-SI) and new LTBI with a long time of incarceration (LTBI-LI), compared to PDL with active TB (ATB), non-infected PDL with a short time of incarceration (NI-SI), and non-infected PDL with a long time of incarceration (NI-LI).

A principal component analysis (PCA) from normalized read count was done to evaluate relationships among samples.

Differential expression was assessed using the DESeq2 package in Bioconductor [35], yielding a result based on a negative binomial model [36]. The differential expression was defined

as a change in the log-fold change (logFC). Only genes with |logFC| >1.0 and an adjusted p-value < 0.1 were considered to be differentially expressed. A log-fold change (logFC) >1.0 means the genes were at least twice expressed (fold change >2.0), and we consider this change could reflect important differences between groups. For different genes, a bigger fold change does not imply a more significant effect on the downstream processes, so a slight change in the gene expression in this population could represent a significant effect.

To identify group- differentially expressed genes, pairwise group comparisons were undertaken; we compared each group with the other four for a total of 10 comparations. After that, we selected the specific genes for each group and analyzed the differences in the enrichment of KEGG pathways using InterMiner (v.1.4.1) [37–39].

## Results

### Participants

All participants were males with current consumption of smoked cocaine derivatives or weed (28.3%), cigarettes (45.2%) and alcohol (27.5%). Only one non-infected individual was underweight (BMI<18.5 kg/m$^2$). Table 1 summarizes demographic information of people with active TB, LTBI and non-infected people. S3 File reports the sociodemographic information for each participant.

### RNA sequencing data

On average, the cDNA libraries generated a read count median 26,362,401; IQR [24,354,000–28,105,266], of 151 nucleotides in length, with Q30 median of 93.6; IQR [93.4–93.8]. After quality trimming, an average of 20 million reads were mapped to GRCh38. Considering all libraries, 34,973 genes were detected, with a median normalized read count of 3.2 [IQR:

**Table 1. Baseline characteristics of people deprived of liberty with active TB, latent tuberculosis infection and non-infected according to the time of incarceration.**

| Variable | ATB | LTBI-SI | LTBI-LI | NI-SI | NI-LI |
|---|---|---|---|---|---|
| | **n = 10** | **n = 8** | **n = 7** | **n = 6** | **n = 5** |
| Age, years, median [IQR] | 33 [24–37] | 34.5 [30.5–39] | 56 [26–62] | 24.5 [22–27] | 39 [35–41] |
| BMI, median [IQR] | 20.8 [20.1–23.4] | 24.0 [19.2–25.7] | 22.9 [20.4–26.7] | 21.3 [18.7–25.6] | 23.7 [21.5–26.3] |
| Time into prison, months, median [IQR] | 34.1 [15–69] | 12.1 [9.9–12.5] | 39.4 [24.2–60.3] | 13.4 [12.7–14.7] | 43.8 [28.7–56.8] |
| At least one comorbidity | 2 | 1 | 2 | 1 | 1 |
| *COPD* | 0 | 1 | 0 | 1 | 0 |
| *Diabetes* | 1 | 0 | 2 | 0 | 0 |
| Current smoke drugs use | 6 | 1 | 0 | 2 | 0 |
| Current inhaled drugs use | 2 | 0 | 0 | 1 | 0 |
| Current cigarettes use | 5 | 4 | 2 | 4 | 1 |
| Current alcohol use | 6 | 2 | 0 | 0 | 1 |
| Presence of BCG scar | 10 | 6 | 5 | 5 | 4 |
| Contact with a TB case | 5 | 0 | 0 | 1 | 1 |
| Prison | | | | | |
| *Prison 1* | 10 | 2 | 7 | 0 | 5 |
| *Prison 2* | 0 | 6 | 0 | 6 | 0 |

LTBI-LI: Latent tuberculosis infection with long incarceration (people already had ≥ 1 year in prison when entering the study). LTBI-SI: Latent tuberculosis infection with short incarceration (people started the follow-up with less than three months of incarceration). ATB: active tuberculosis. NI-SI: non-infected with short incarceration. NI-LI: non-infected with long incarceration. IQR: interquartile range. TB: Tuberculosis. BMI: Body mass index (kg/m$^2$). COPD: Chronic obstructive pulmonary disease. BCG: bacille Calmette-Guerin

0–130.8]. The PCA from normalized read count revealed evidence for intragroup heterogeneity (S1 Fig).

## Differential gene expression among the groups (LTBI, ATB and NI)

The initial comparison involving PDL with ATB, LTBI and NI showed few or null differentially expressed genes (DEGs) among the three groups. People with LTBI had one up-regulated (METRNL) and one down-regulated gene (unknown gene) compared to non-infected participants. We observed one up-regulated gene when we compared the LTBI with the ATB group (MAFF interacting protein), and there were no DEGs among the ATB and non-infected people (S1 Table).

## Differential gene expression among the groups, considering the time of incarceration

The pairwise comparisons among the five groups showed 600 DEGs, with 298 up-regulated and 302 down-regulated genes. These differentiated genes were not previously noted, showing that in this population, the time of incarceration is a variable that should be included in the analysis.

The higher gene expression was found in the PDL with LTBI-SI, with 235 genes (119 up-regulated and 116 down-regulated genes) compared to NI-LI, and 175 genes (61 up-regulated and 114 down-regulated genes) when compared with the LTBI-LI group. Also, in the LTBI-SI group, there were 18 DEGs compared to the NI-SI and one DEG with the ATB group.

PDL with LTBI-LI displayed 23 DEGs (13 up-regulated and 10 down-regulated) compared to PDL with ATB. Conversely, when the LTBI-LI group was compared to NI-LI, 49 DEGs were identified (47 up-regulated and 2 down-regulated). When they were compared with PDL with NI-SI, 5 DEGs were identified (4 up-regulated and 1 down-regulated gene).

When the NI-LI was compared to the ATB group, 25 DEGs were identified, 20 down-regulated and five up-regulated. No differentially expressed genes were detected between NI-SI and ATB groups (S1 Table).

## Specific and differential gene expression among the five groups

There were 417 genes differentially expressed when comparing the five groups altogether. Most of the differentially expressed genes were observed in the LTBI-SI group (with 126 and 163 genes up and down-regulated, respectively), and the fewest in the ATB group (22 and 4 genes up and down-regulated). Fig 2 depicts the Venn diagrams for specific up or down-regulated genes. The S2 Table reports the upregulated and downregulated genes exclusively found in each group with the ID number, symbol, functional description, and if that gene has been found in previous publications.

## Pathway analysis

When we evaluated the enrichment of pathways using the differentially expressed genes for each group, in individuals with LTBI-LI, the only enriched pathway within up-regulated genes was Emi1 phosphorylation, and no enriched pathways were detected using the down-regulated genes. No enriched pathway was detected among specific differentially expressed genes in people with active TB.

In individuals with LTBI-SI, there were three pathways accounted for the highest number of up-regulated genes: cytokine signaling, signal transduction, and the immune system, and

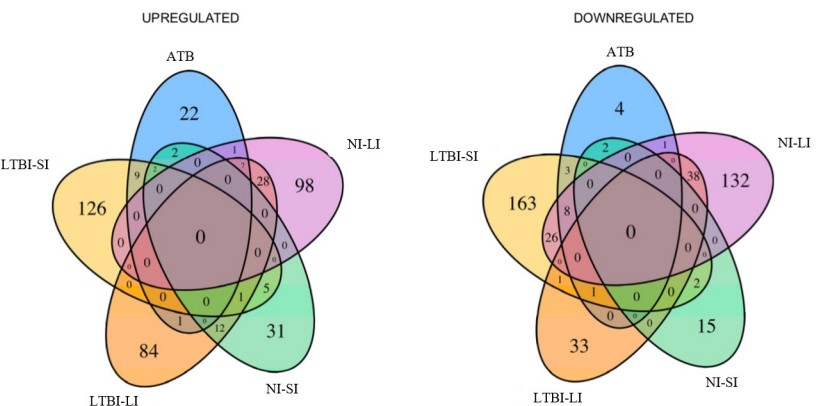

**Fig 2. Number of specific differentially expressed genes (up or down-regulated) between new latent tuberculosis infection, active tuberculosis TB and non-infected individuals.** LTBI-LI: Latent tuberculosis infection with long incarceration (people already had ≥ 1 year in prison when entering the study). LTBI-SI: Latent tuberculosis infection with short incarceration (people started the follow-up with less than three months of incarceration). ATB: active tuberculosis. NI-SI: non-infected with short incarceration. NI-LI: non-infected with long incarceration.

there were two pathways involving a larger number of down-regulated genes: neutrophil degranulation and the innate immune system (Fig 3).

On the other hand, the differential expression of genes related to the cell cycle was observed in non-infected individuals. Those pathways were detected among upregulated genes in the NI-SI group (Fig 4), and among down-regulated genes in the NI-LI group (Fig 5). In addition, pathways related to the immune system were observed in the NI-LI group (Fig 5).

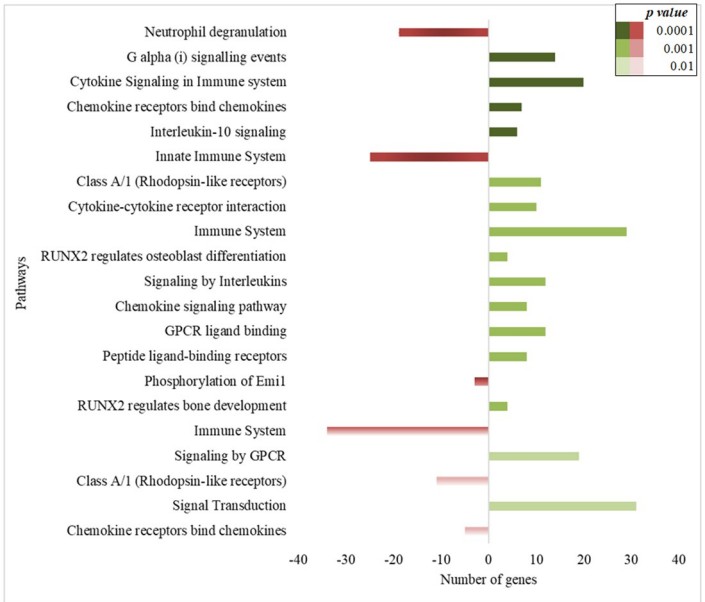

**Fig 3. Enrichment pathways analysis using the differentially expressed genes in people with new latent tuberculosis infection and with short incarceration (LTBI-SI).** A short time of incarceration means people who started the follow-up with less than three months of incarceration. The negative and positive numbers show the number of down-regulated (red bars) and up-regulated genes (green bars) in each pathway related to the biological process.

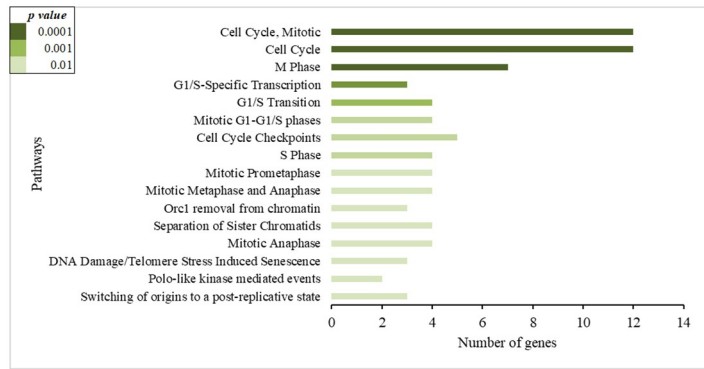

**Fig 4. Enrichment pathways analysis using the differentially expressed genes in people non-infected by** *Mycobacterium tuberculosis* **and with short incarceration (NI-SI).** A short time of incarceration means people who started the follow-up with less than three months of incarceration. The bars show the number of up-regulated genes in each pathway related to the biological process.

## Discussion

Most studies that have evaluated differential gene expression from clinical samples have attempted to predict the risk of progressing to active TB if infected with MTB. To our knowledge, this is one of the first studies to identify biosignatures in people who recently converted their TST (new LTBI) among non-infected individuals.

This study found that: 1. In an environment with a high prevalence of active TB and high exposure to MTB, the time of incarceration influence the DEGs, and there are different gene

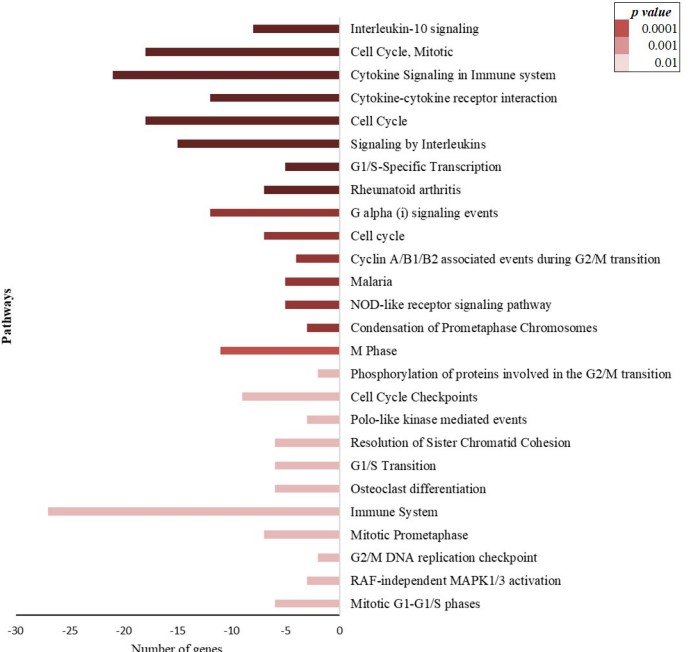

**Fig 5. Enrichment pathways analysis using the differentially expressed genes in people non-infected by** *M tuberculosis* **and with long incarceration (NI-LI).** A long time of incarceration means people who already had ≥ 1 year in prison when entering the study. Bars show the number of down-regulated genes in each pathway related to the biological process.

expression profiles between people with short and long incarceration that did or did not become infected, compared to people with active TB. 2. The cellular processes or pathways reported vary among groups, mainly related to the immune system or the cell cycle.

Recent studies not related to tuberculosis have shown that the analysis stratification using biological variables like sex [40] and genotypes [41] influence the gene expression profile. Our study shows the effect of an external variable on the gene expression profile in a population with a high risk of infection.

Our results showed that in non-infected individuals that have been incarcerated for more than a year (NI-LI), the differentially expressed genes are predominantly related to processes involved in the cell cycle. For many intracellular bacterial pathogens manipulating the host cell is a common strategy to promote infection. A study by Cumming *et al.* showed that, in macrophages infected by MTB, the bacterium regulates and arrests the cell cycle, and also showed that seven of ten host pathways discovered were involved in the host's cell cycle, and these pathways including DNA damage checkpoints, biomechanical stress, cytoskeletal remodeling, chromosome condensation and apoptosis [42]. We hypothesized that individuals who are persistently TST negative have some mechanisms related to the cell cycle that prevents MTB infection and subsequent regulation of the cell cycle.

Another study conducted in Uganda that used genome-wide transcriptional profiles from stimulated monocytes with MTB, of household contacts of patients with TB that remained TST-negative after two years of having contact with the TB case, showed that the most significant pathway in the persistently negative TST people was the histone deacetylase (HDAC) [43], an important molecule for the immune response to in vitro human macrophages and in vivo zefrafish models of MTB infection [44]. We did not find any pathway related to histone deacetylase, but we found two genes expressed in non-infected groups and related to histone clusters: H4C8 and H2AC8. The Uganda study also found decreased gene expression of the NOD-like receptors (NLRs) signaling pathway; similar to them, we found five downregulated genes involved in this pathway in NI-LI individuals. The NLRs recognize ligands from microbial pathogens, and it has four general functions: inflammasome formation, signaling transduction, transcription activation, and autophagy [45]. NOD2 is a receptor for bacterial peptidoglycans and participates in recognizing mycobacteria [46]. These data suggest that future studies of undelaying the dynamics between the bacterium and the host as they relate to the NOD-like receptor (NLRs) and histone pathways could be fruitful in identifying novel drug or vaccine strategies [47].

During follow-up, most people in the NI-LI group had more than three years of incarceration and remained persistently TST negative despite being in an environment with high exposure to MTB. This phenomenon has been previously reported, and 7% to 35% of individuals may be 'resistant' to MTB infection or exhibit "early clearance". These individuals will remain TST or IGRA negative despite heavy and continued exposure to MTB and are not at risk for progression to active TB [48]. Some hypotheses that could explain this phenomenon are: the individuals immediately clear the bacteria at the site of infection due to a robust innate immune response without the stimulation of an acquired immune response [49], or have a complete innate 'resistance' to infection and disease due to gene variants, such as those contained in the genes Toll-like receptor-4 [50], ZEB2 and GTDC1 [51] genes. Another factor that has been described as associated with resistance to MTB infection is the long-term cohabitation of some populations with the mycobacterium. This hypothesis suggests that the lack of exposure to a pathogen leads to hyper-susceptibility to infection [52] and might explain why people recently incarcerated have a higher risk of becoming infected. We acknowledge the limitation that TST is an imperfect diagnostic tool for LTBI, and the T cell anergy may cause false negative results on the TST test [48].

In people with a new LTBI-SI, our results showed that many differentially expressed genes are involved in pathways related to the immune processes. Upregulated genes were primarily involved in Cytokine Signaling pathways, Immune system, Signaling by G Protein-Coupled receptor (GPCR), and Signal Transduction. In contrast, downregulated genes were related to the Neutrophil degranulation and Innate Immune System pathways. It is appreciated that upon exposure to MTB, the outcome of infection (clearance, LTBI, ATB) is determined by an immune response that relies on the participation of diverse innate and adaptive cells, including macrophages, dendritic cells, T cells, and neutrophils [53, 54]. Similarly, studies that evaluated the differential expression between LTBI, active TB and uninfected individuals reported the participation of pathways related to the immune system in response to infection or tuberculosis disease, such as regulation of leukocytes, B cell and lymphocyte-mediated response [55], and interferon-gamma signaling [56]. Likewise, it has been reported that GPCR expression is elevated at both mRNA and protein levels in macrophages in response to BCG infection, and it is essential for the entry of MTB into these cells. Therefore, according to our results, it is consistent that this pathway is increased in individuals with early MTB infection [57].

Some genes found in our cohort study in people with LTBI-SI (FCGR3B, CXCR1, OASL, LRRC32 and FAM157C) and NI-LI (CXCR1, MSRB1, OR52K3P) have been reported previously by Weng Kwan P *et al*. [58]. They studied exposed household contacts of TB patients and compared the expression with healthy volunteers without a recent history of TB exposure. They identified a 186-gene signature that can differentiate people with recent exposure to TB from those without recent exposure, with a higher risk score in contacts that were IGRA positive versus IGRA negative. These results support the idea that the time of exposure to a TB case is a variable that plays an important role in the gene expression dynamics. In prisons or other environments with a high burden of TB, being in contact with a case may go unnoticed. Still, other factors, such as overcrowding and long stays in closed places, make frequent exposure to mycobacteria possible. As a result, they can have either of the two outcomes, become infected or remain uninfected, according to the potential reasons discussed above.

In active TB cases, we found that of the 26 genes differentially expressed, six were related to some immune function: TNF receptor superfamily, interferon-induced protein, T cell variable delta receptor, and CD24 molecule. Three studies have previously reported the TNF receptor superfamily genes when they evaluated the expression from stimulated and unstimulated PBMCs, and from whole blood samples, and whose expression can differentiate patients with active TB from individuals with LTBI [55, 56, 59]. We also found three previously reported genes that were able to distinguish active TB from LTBI: TMED7, MPO and KCNMA1. Walter ND, *et al*. previously reported that expression of TMED7 combined with 50 or 118 other transcripts obtained by microarrays expression analysis is able to distinguish active TB from LTBI, or a combined group of pneumonia and LTBI [60]. This gene is related to pathways of metabolism of proteins and Toll-like Receptor Signaling [61]. MPO (Myeloperoxidase) was first reported to be involved in LTBI by Kaforou M., *et al*. [62]. MPO is an enzyme present in the lysosomes of monocytes and neutrophils, and it catalyzes the formation of reactive oxygen intermediates, including hypochlorous, hypobromous, and hypothiocyanous acids [63], and has been implicated in TB response [64, 65]. Finally, the KCNMA1 gene (potassium calcium-activated channel subfamily M alpha 1) was recently reported by Tabone O., *et al*. [23] to be involved in the clinical TB stage, and was also related to TB *in vivo* studies of gene expression of lung granulomas isolated from sham-vaccinated nonhuman primates at 10 weeks after infection with *M. tuberculosis* [66].

Even though there are several strategies for the identification of active TB using RNA sequencing, two approaches have been extensively studied: the first is to identify people with active TB compared with LTBI, and the second is to identify individuals with LTBI who have a

higher risk of progressing to active TB. The main limitation when conducting studies aimed at differentiating patients with active TB from people with latent infection is that LTBI has been described as a highly heterogeneous state, where some individuals may have lesions similar to those of active TB at the lung level, without symptoms, and with slow or null progression, while others are limited to having a chronic non-progressive infection [67, 68]. The spectrum of individuals included in the LTBI groups may have included a few persons with subclinical TB- terms that were conceptualized after the inception and conduct of this study [68]. Also, it is important to highlight that the differences in the results between our study and others may be partially explained by different genetic backgrounds in the populations, as previous studies were conducted in South India [17], the United States [19], Peru [19], African countries [14], China [18], among others. Therefore, we advocate for multicenter research to combine and compare different populations worldwide.

The main strength of this study is the access to samples that enable the detection of RNA expression signatures in people recently infected with MTB with different times of incarceration (a proxy to the time of exposure to MTB), who did not develop active TB, at least in the 12 months following conversion. These results are novel because the identification of specific biomarkers of new MTB infection will be crucial to develop LTBI diagnostic tests for early diagnosis in a group that is at the highest risk for progression to active TB, and for whom offering preventive therapy can decrease the progression to active TB, and the number of people with infectious TB, therefore improving TB control.

A potential limitation of our study is that it has been described that gene expression is influenced by genetic ancestry, but the Colombian population is known to be admixed [69], and the ancestry effect could be similar among PDL. Another limitation is that the group of PDL with long incarceration was older than the other groups. Age can affect gene expression from blood samples [70], but our sample size is too small to test the hypothesis that changes are related to age in converters or exposed people, or both. Studies with larger sample sizes powered to perform multivariable analyses are necessary to address this limitation.

A final consideration is a potential discrepancy in TST positivity which may be associated with cross-reactive antigens responses. The main concerns about the TST cross-reactivity are the BCG vaccination and infection with non-tuberculous mycobacteria. In Colombia, the BCG vaccination is administered at birth. The BCG vaccine administered at birth has a minimum effect on TST specificity, especially if the TST is administered ten years or more after vaccination [71]. We consider that this does not affect our study's results because all the participants were 18 years old or more at baseline [72].

Similarly, Colombia does not have a high prevalence of non-tuberculosis mycobacteria (NTM). In Medellin, most non-tuberculosis mycobacteria are present in people with immunosuppression. A Colombian research conducted from 2004 to 2011 in a tertiary hospital reported that among people with a positive mycobacterial culture, most people were immunosuppressed (HIV, chronic use of steroids or immunosuppressants, chronic kidney disease, diabetes mellitus, etc.). The frequency of NTM was 9.1% [73]. In the prisons where the participants were incarcerated, there were no reports of NTM in our group's previous study of tuberculosis disease [24].

In conclusion, we found several differentially expressed genes between LTBI and non-infected individuals that have different times of exposure (recently incarcerated and with one or more years of incarceration at the time of starting the follow-up) compared to active TB. Future studies should combine data from international consortia to allow further comparisons between populations with different genetic backgrounds and encourage longitudinal studies that will enable improved understanding of new infections and the early interactions between *M. tuberculosis* and the host.

## Supporting information

**S1 Fig. Principal component analysis among the five groups, to evaluate the heterogeneity.**
LTBI-LI: Latent tuberculosis infection with long incarceration (people already had ≥ 1 year in prison when entering the study). LTBI-SI: Latent tuberculosis infection with short incarceration (people started the follow-up with less than three months of incarceration). ATB: active tuberculosis. NI-SI: non-infected with short incarceration. NI-LI: non-infected with long incarceration.
(TIF)

**S1 Table. Differential gene expression among each group; this table shows the results from the pairwise comparisons among the 5 groups.** LTBI-LI: Latent tuberculosis infection with long incarceration (people already had ≥ 1 year in prison when entering the study). LTBI-SI: Latent tuberculosis infection with short incarceration (people started the follow-up with less than three months of incarceration). ATB: active tuberculosis. NI-SI: non-infected with short incarceration. NI-LI: non-infected with long incarceration.
(PDF)

**S2 Table. Specific differential expression in each group.** LTBI-LI: Latent tuberculosis infection with long incarceration (people already had ≥ 1 year in prison when entering the study). LTBI-SI: Latent tuberculosis infection with short incarceration (people started the follow-up with less than three months of incarceration). ATB: active tuberculosis. NI-SI: non-infected with short incarceration. NI-LI: non-infected with long incarceration.
(PDF)

**S1 File. PLOS' questionnaire on inclusivity in global research.**
(PDF)

**S2 File. Selection criteria and definitions used during the cohort study.** Sep 2016-dec 2018, Colombia.
(PDF)

**S3 File. Database including the participants' sociodemographic information.** (Excel, sheet 1); Excel, sheet 2 shows the definition of variables.
(XLSX)

## Acknowledgments

The authors are grateful to all of the PDL who accepted to participate in the study; to INPEC (Instituto Nacional Penitenciario y Carcelario de Colombia), and the director of each prison and all personnel working there for their support in performing the study. The field and laboratory team for their work and collaboration during the study period.

## Author Contributions

**Conceptualization:** Mariana Herrera, Yoav Keynan, Paul J. McLaren, Lucelly López, Diana Marin, Zulma Vanessa Rueda.

**Data curation:** Paul J. McLaren, Juan Pablo Isaza.

**Formal analysis:** Mariana Herrera, Paul J. McLaren, Juan Pablo Isaza, Bernard Abrenica, Lucelly López.

**Funding acquisition:** Diana Marin, Zulma Vanessa Rueda.

**Investigation:** Mariana Herrera, Yoav Keynan, Zulma Vanessa Rueda.

**Methodology:** Paul J. McLaren, Juan Pablo Isaza, Bernard Abrenica, Zulma Vanessa Rueda.

**Project administration:** Yoav Keynan, Lucelly López, Diana Marin, Zulma Vanessa Rueda.

**Software:** Juan Pablo Isaza, Bernard Abrenica.

**Supervision:** Yoav Keynan, Zulma Vanessa Rueda.

**Writing – review & editing:** Mariana Herrera, Yoav Keynan, Paul J. McLaren, Juan Pablo Isaza, Lucelly López, Diana Marin, Zulma Vanessa Rueda.

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
