## [Decision Letter · Decision Letter 0]

10 Jun 2022

PONE-D-22-09296Gene expression profiling identifies candidate biomarkers for new latent tuberculosis infections. A cohort studyPLOS ONE

Dear Dr. Rueda,

Thank you for submitting your manuscript to PLOS ONE. After careful consideration, we feel that it has merit but does not fully meet PLOS ONE’s publication criteria as it currently stands. Therefore, we invite you to submit a revised version of the manuscript that addresses the points raised during the review process. I would recommend re-analyzing the data and making the data available to reviewers for better understanding and acceptance of the paper. Kindly see the comments by Reviewer 1 and 2 which highlight issues with the manuscript. If changes are made as requested, the manuscript is of value and can be accepted for publication.

We look forward to receiving your revised manuscript.

Kind regards,

Afsheen Raza, PhD

Academic Editor

PLOS ONE

Journal Requirements:

2. Please provide additional information regarding the considerations made for the prisoners included in this study. For instance, please discuss whether participants were able to opt out of the study and whether individuals who did not participate receive the same treatment offered to participants.

3. Please include a complete copy of PLOS’ questionnaire on inclusivity in global research in your revised manuscript. Our policy for research in this area aims to improve transparency in the reporting of research performed outside of researchers’ own country or community. The policy applies to researchers who have travelled to a different country to conduct research, research with Indigenous populations or their lands, and research on cultural artefacts. The questionnaire can also be requested at the journal’s discretion for any other submissions, even if these conditions are not met.  Please find more information on the policy and a link to download a blank copy of the questionnaire here: https://journals.plos.org/plosone/s/best-practices-in-research-reporting. Please upload a completed version of your questionnaire as Supporting Information when you resubmit your manuscript.

Reviewers' comments:

Reviewer's Responses to Questions

**Comments to the Author**

1. Is the manuscript technically sound, and do the data support the conclusions?

Reviewer #1: No

Reviewer #2: Yes

Reviewer #3: No

2. Has the statistical analysis been performed appropriately and rigorously? 

Reviewer #1: I Don't Know

Reviewer #2: Yes

Reviewer #3: No

3. Have the authors made all data underlying the findings in their manuscript fully available?

Reviewer #1: No

Reviewer #2: No

Reviewer #3: Yes

4. Is the manuscript presented in an intelligible fashion and written in standard English?

Reviewer #1: No

Reviewer #2: Yes

Reviewer #3: Yes

5. Review Comments to the Author

Reviewer #1: This is a gene expression study.

However, the author haven't made their data available (in particular the raw gene expression matrix) for us reviewers. The submitted manuscript states: "Data will be made publicly available in a public, open access repository in case of

acceptance of the paper".

Therefore, I have no way of judging whether the data analysis their performed is sound and if the conclusions of the manuscript are correct.

Reviewer #2: In this manuscript, the authors explored differential expression profiles for individuals with newly latent tuberculosis infection (LTBI), active tuberculosis (TB) and those exposed to TB but not infected. The research is very interesting and meaningful, although the bioinformatic analysis is quite simple. I would suggest the authors to address the following minor questions to improve the manuscript:

1) The authors divided non-infected people and new TST converters into SI and LI groups according to the incarceration time. Here, the time into prison is a very important characteristic throughout the analysis. Although the authors listed the median of incarceration time in the table of baseline characteristics, the authors should display the incarceration time of every individual in a supplementary figure or table.

2) The authors respectively described differential gene expression profiles among SI and TI groups (i.e., LTBI-SI vs NI-SI, LTBI-LI vs NI-LI,). Is there any difference between SI and TI groups (i.e., LTBI-SI vs LTBI-LI, NI-SI vs NI-LI,)? The authors should mention it in the manuscript.

3) The authors also compared the five groups to perform differential gene expression and pathway analyses. Does it mean that the authors compare one group with other four ones, and repeat this process for each group? The author should clearly state the comparisons for this part in the MS.

Reviewer #3: The identification of gene biomarkers associated with latent TB infection and those associated with risk of disease are important. The study is of interest but in its current state it cannot be clearly evaluated correctly.

Regarding the gene expression analysis, the authors state “The differential expression was defined as a change in the log-fold change (logFC). Only genes with |logFC| >1.0 and an adjusted p-value < 0.1 were considered to be differentially expressed’

These cut-off are rather low. Log Fold change of > or < 1.5 or 2 should be considered together with a adjusted p-value of 0.05 to be statistically significant. Given the analysis cut offs used, it is difficult to understand the study results and these should be reanalyzed.

Further, the group sizes are small and by dividing the TST converters into TST-SI and TST-LI, the robustness of the comparison between TST converters and TST negative individuals is not apparent.

The authors are advised to run a primary analysis on the group without stratification into SI and LI, and compare TST converters and non-converters with individuals with Active TB.

The authors need to discuss the possible discrepancy in TST positivity which may be associated with cross reactive antigens responses.

What is the incidence of TB in Columbia? What is the rate of latent TB infection in the population – if this is known.

The results need to be written in a more comprehensive manner in the context of the revised results

Figure legends need to be written appropriately describing the data, defining acronyms and the analysis shown.

6. PLOS authors have the option to publish the peer review history of their article (what does this mean?). If published, this will include your full peer review and any attached files.

Reviewer #1: No

Reviewer #2: No

Reviewer #3: No

---

## [Author Response · Author response to Decision Letter 0]

22 Aug 2022

Reviewers' comments:

Reviewer #1:

• This is a gene expression study. However, the author haven't made their data available (in particular the raw gene expression matrix) for us reviewers. The submitted manuscript states: "Data will be made publicly available in a public, open access repository in case of acceptance of the paper". Therefore, I have no way of judging whether the data analysis their performed is sound and if the conclusions of the manuscript are correct.

Answer: Thanks to the reviewer for their recommendation. The raw sequences were uploaded to the Sequence Read Archive (SRA) data repository, BioProject Id: PRJNA858854.

Reviewer #2:

• In this manuscript, the authors explored differential expression profiles for individuals with newly latent tuberculosis infection (LTBI), active tuberculosis (TB) and those exposed to TB but not infected. The research is very interesting and meaningful, although the bioinformatic analysis is quite simple. I would suggest the authors to address the following minor questions to improve the manuscript:

Answer: We thank the reviewer for the kind words regarding the impact of our work. We feel the simplicity of the analysis is a strength as we show the differentially expressed genes effectively discriminate between the groups under study without the need for more sophisticated models. The simplicity of our method also allows it to be broadly applied in multiple contexts providing the opportunity for direct comparison between ours and future studies. 

• The authors divided non-infected people and new TST converters into SI and LI groups according to the incarceration time. Here, the time in prison is a very important characteristic throughout the analysis. Although the authors listed the median of incarceration time in the table of baseline characteristics, the authors should display the incarceration time of every individual in a supplementary figure or table.

Answer: Thanks to the reviewer for this recommendation. We support open science, and we included as supplementary material the incarceration time for every individual, as you suggested, and also additional sociodemographic characteristics of the participants. Each participant has an internal ID code and the information we collected during the research. The data is reported in the S3 file.

• The authors respectively described differential gene expression profiles among SI and TI groups (i.e., LTBI-SI vs NI-SI, LTBI-LI vs NI-LI,). Is there any difference between SI and TI groups (i.e., LTBI-SI vs LTBI-LI, NI-SI vs NI-LI,)? The authors should mention it in the manuscript.

Answer: we agree with the reviewer; we omitted this information, which is relevant to the article. Now, we have included this information in the manuscript. Also, in the supplementary material, the reviewer can see the information you asked for and a summary of the number of genes differentially expressed (Des) in each group. The table below, also included in the supplementary material, shows the summary.

 Differentially Expressed Genes 

Contrast Total Up Down 

LTBI-SI_vs_NI-LI 235 119 116

LTBI-SI_vs_NI-SI 1 1 0

LTBI-SI_vs_ATB 18 1 17

LTBI-SI_vs_LTBI-LI 175 61 114

LTBI-LI_vs_NI-SI 5 1 4

LTBI-LI_vs_NI-LI 49 47 2

LTBI-LI_vs_ATB 23 13 10

NI-SI_vs_NI-LI 69 50 19

NI-SI_vs_ATB 0 0 0

NI-LI_vs_ATB 25 5 20

Total * 600 298 302

* Some genes could be in different categories

• The authors also compared the five groups to perform differential gene expression and pathway analyses. Does it mean that the authors compare one group with other four ones, and repeat this process for each group? The author should clearly state the comparisons for this part in the MS.

Answer: We agree with the reviewer. Considering their recommendation, we have updated the analysis section in the manuscript. 

Reviewer #3:

• The identification of gene biomarkers associated with latent TB infection and those associated with risk of disease are important. The study is of interest but in its current state it cannot be clearly evaluated correctly.

Regarding the gene expression analysis, the authors state “The differential expression was defined as a change in the log-fold change (logFC). Only genes with |logFC| >1.0 and an adjusted p-value < 0.1 were considered to be differentially expressed’

These cut-offs are rather low. Log Fold change of > or < 1.5 or 2 should be considered together with a adjusted p-value of 0.05 to be statistically significant. Given the analysis cut offs used, it is difficult to understand the study results and these should be reanalyzed.

Answer: We thank the reviewer for raising this point, and we would like to explain why our decision is related to the cut-off for the differential gene expression. First, log-fold change (logFC) >1.0 means the genes were at least twice expressed (fold change >2.0), and we consider this change could reflect important differences between the groups. For different genes, a bigger fold change does not imply a more significant effect on the downstream processes, so a slight change in the gene expression in this specific population could represent a significant effect. 

On the other hand, considering that evaluation of gene expression profiles of new infections by M. tuberculosis is not common in previously published articles and our paper is one of the first approaches, we preferred to be less stringent in detecting the differential gene expression, trying to extract the relevant biological signals from the groups. This information also explains why we chose an adjusted p-value < 0.1.

• Furthermore, the group sizes are small and by dividing the TST converters into TST-SI and TST-LI, the robustness of the comparison between TST converters and TST negative individuals is not apparent.

The authors are advised to run a primary analysis on the group without stratification into SI and LI, and compare TST converters and non-converters with individuals with active TB.

Answer: We agree with the reviewer. We did the primary analysis and found three differential expressed genes among the newly infected people, patients with active TB, and those who remained with a negative TST during the follow-up. Initially, we did not show these results to keep the manuscript as simple as possible. After your suggestion, we have included both results in the manuscript, and also, we have included information related to them in the discussion and supplementary sections. 

Regarding the time of incarceration, in our previous epidemiological [1] and immunological analysis (in preparation for publication), using the same population, we found differences in the risk of getting a new infection for M. tuberculosis and the concentration of the 12 immune parameters according to their incarceration time (short versus a long time of incarceration). In prisons, our group found that the “tuberculosis infection incidence rate varies between 2,403 cases per 100,000 person-months (95% CI 1,364.62-4,231.10) in PDL with a short time of incarceration (those who were enrolled in follow-up upon incarceration or within the first 3 months of incarceration), to 419.66 cases per 100,000 person-months (95% CI 225.80-779.95) in PDL with a long time of incarceration (individuals who started their follow-up after 1 year or more of incarceration)[1].” Therefore, we decided to report the results stratified by this variable. We consider the results showing an external variable’s influence on the gene expression profile is important to discuss and to be included in future research and analysis. 

Regarding the group size, some studies have shown that using a fold-change = 2, three replicates per condition are enough to detect the differential expressed genes[2]. The cited paper also mentions that the ideal scenario is to have at least six replicates per condition for all experiments. All of our groups are bigger than six, except the non-infected with a long time of incarceration group (n=5). 

• The authors need to discuss the possible discrepancy in TST positivity which may be associated with cross reactive antigens responses.

Answer: Thanks to the reviewer for this relevant comment; we agree that cross-reactivity is a topic that all the studies related to tuberculosis infection need to consider. We included in the discussion the following paragraphs: 

The main concerns about the TST cross-reactivity are the BCG vaccination and infection with non-tuberculous mycobacteria. In Colombia, the BCG vaccination is administered at birth. The BCG vaccine administered at birth has a minimum effect on TST specificity, especially if the TST is administered ten years or more after vaccination[3]. We consider that this does not affect our study’s results because all the participants were 18 years old or more at baseline[4].

Similarly, Colombia does not have a high prevalence of non-tuberculosis mycobacteria (NTM). In Medellin, most non-tuberculosis mycobacteria are present in people with immunosuppression. A colombian research, conducted from 2004 to 2011 in a tertiary hospital reported that among people with a positive mycobacterial culture, most people were immunosuppressed (HIV, chronic use of steroids or immunosuppressants, chronic kidney disease, diabetes mellitus, etc.). The frequency of NTM was 9.1%, [5]. In the prisons where the participants were incarcerated, there were no reports of NTM in our group's previous study of tuberculosis disease[6]. 

• What is the incidence of TB in Columbia? What is the rate of latent TB infection in the population – if this is known. 

Answer: The TB incidence in Colombia is 22 cases per 100,000 inhabitants[7] [REF]. Our group found in 2010 to 2012, that the TB incidence in four prisons in Colombia was 505 cases per 100,000 [6].

In prisons, our group also found in four cohorts (2134 PDL that were investigated to rule out TB and 240 PDL with two-step TST negative and followed them to evaluate TST conversion) that “tuberculosis infection incidence rate varies between 2,403 cases per 100,000 person-months (95% CI 1,364.62-4,231.10) in PDL with short time of incarceration (less than three months of incarceration at baseline), to 419.66 cases per 100,000 person-months (95% CI 225.80-779.95) in PDL with long time of incarceration (individuals who were enrolled for the follow after at least 1 year of incarceration) [1].”

The participants that we included for the gene expression profile evaluation come from a cohort study conducted between 2016 - 2018. 

We included this information and reference into the manuscript (Study design and Population section).

• The results need to be written in a more comprehensive manner in the context of the revised results

Answer: Thanks, you were right. We have reviewed and worked on the results to improve them.

• Figure legends need to be written appropriately describing the data, defining acronyms and the analysis shown.

Answer: Thanks, you were right. We have reviewed the description of the data, the acronyms and the analysis shown in each figure, table and supplementary material. 

 Answer: We have processed our figure files using PACE and the we added to the submission the version generated by PACE. 

Thanks again for your kind review and comments.

All authors

---

## [Editor Report · Decision Letter 1]

25 Aug 2022

Gene expression profiling identifies candidate biomarkers for new latent tuberculosis infections. A cohort study

PONE-D-22-09296R1

Dear Dr. Zulma,

We’re pleased to inform you that your manuscript has been judged scientifically suitable for publication and will be formally accepted for publication once it meets all outstanding technical requirements.

Kind regards,

Afsheen Raza, PhD

Academic Editor

PLOS ONE
---

## [Editor Report · Acceptance letter]

19 Sep 2022

PONE-D-22-09296R1 

Gene expression profiling identifies candidate biomarkers for new latent tuberculosis infections. A cohort study 

Dear Dr. Rueda:

I'm pleased to inform you that your manuscript has been deemed suitable for publication in PLOS ONE. Congratulations! Your manuscript is now with our production department. 

Kind regards, 

on behalf of

Dr. Afsheen Raza 

Academic Editor

PLOS ONE